# Learning filter widths of spectral decompositions with wavelets

**Haidar Khan**
Department of Computer Science
Rensselaer Polytechnic Institute
Troy, NY 12180
khanh2@rpi.edu

**Bülent Yener**
Department of Computer Science
Rensselaer Polytechnic Institute
Troy, NY 12180
yener@rpi.edu

## Abstract

Time series classification using deep neural networks, such as convolutional neural networks (CNN), operate on the spectral decomposition of the time series computed using a preprocessing step. This step can include a large number of hyperparameters, such as window length, filter widths, and filter shapes, each with a range of possible values that must be chosen using time and data intensive cross-validation procedures. We propose the wavelet deconvolution (WD) layer as an efficient alternative to this preprocessing step that eliminates a significant number of hyperparameters. The WD layer uses wavelet functions with adjustable scale parameters to learn the spectral decomposition directly from the signal. Using backpropagation, we show the scale parameters can be optimized with gradient descent. Furthermore, the WD layer adds interpretability to the learned time series classifier by exploiting the properties of the wavelet transform. In our experiments, we show that the WD layer can automatically extract the frequency content used to generate a dataset. The WD layer combined with a CNN applied to the phone recognition task on the TIMIT database achieves a phone error rate of 18.1%, a relative improvement of 4% over the baseline CNN. Experiments on a dataset where engineered features are not available showed WD+CNN is the best performing method. Our results show that the WD layer can improve neural network based time series classifiers both in accuracy and interpretability by learning directly from the input signal.

## 1   Introduction

The spectral decomposition of signals plays an integral role in problems involving time series classification or prediction using machine learning. Effective spectral decomposition requires knowledge about the relevant frequency ranges present in an input signal. Since this information is usually unknown, it is encoded as a set of hyperparameters that are hand-tuned for the problem of interest.

This approach can be summarized by the application of filters to a signal and transformation to the time/frequency domain in a preprocessing step with the short-time Fourier transform (STFT) [27], wavelet transform [10], or empirical mode decomposition [16]. The resulting time series of frequency components is then used for the classification or prediction task. Examples of this approach are present across the spectrum of problems involving time series, including financial time series prediction [7], automatic speech recognition [41, 2, 38], and biological time series analysis [4, 24].

As the parameters of a spectral decomposition are important for time-series problems and are generally not transferable, it is useful to develop methods to efficiently optimize the parameters for each problem. Currently these methods are dominated by cross-validation procedures which incur heavy costs in both computation time and data when used to optimize a large set of hyperparameters.

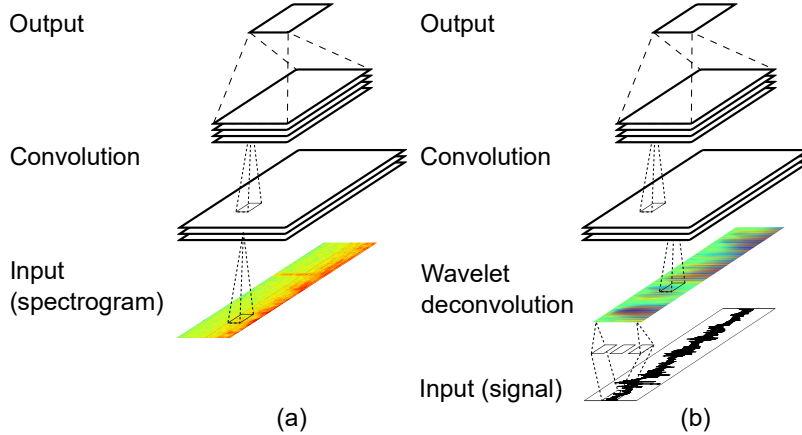

Figure 1: (a) The typical setup in a signal classification problem using deep neural networks with convolution and max-pooling layers applied to a preprocessed spectrogram followed by fully connected classification layers. (b) The proposed setup where the convolutional network operates directly on the input signal via the wavelet deconvolution layer, eliminating the preprocessing step and associated hyperparameters and learning the spectral decomposition using gradient descent. This is accomplished by convolving the input signal and a set of wavelet filters with learnable scales.

We propose a method to efficiently optimize the parameters of the spectral decomposition based on the wavelet transform in a neural network framework.

Our proposed method, called the wavelet deconvolution (WD) layer, learns the spectral decomposition relevant to the classification task with backpropagation and gradient descent. This approach results in a reduction in hyperparameters and a model that is interpretable using properties of the wavelet transform.

This rest of this paper is organized as follows. In Section 2, we introduce the wavelet transform and relevant background. In Section 3, we describe the wavelet deconvolution layer and show how the scale parameters can be learned with gradient descent. Section 4 covers related work. We present our experiments and results in Section 5. We conclude in Section 7.

## 2   Wavelet transform

Given a signal $x(t)$ defined over $t = 1...T$, we begin by describing the continuous wavelet transform (CWT) of the signal [14, 30]. The transform is defined by the choice of a mother wavelet function $\Psi$ that is scaled to form a set of wavelet functions, each of which is convolved with the signal. The mother wavelet function is chosen such that it has small local support and satisfies the zero mean and normalization properties [10]:

$$\int_{-\infty}^{\infty} \psi(t)dt = 0$$

$$|\psi(t)|^2 = \int_{-\infty}^{\infty} \psi(t)\psi^*(t)dt = 1$$

A common choice of mother wavelet function, called the Ricker or Mexican Hat wavelet, is given by the second derivative of a Gaussian:

$$\psi(t) = \frac{2}{\pi^{1/4}\sqrt{3\sigma}}\left(\frac{t^2}{\sigma^2} - 1\right)e^{-\frac{t^2}{\sigma^2}}$$

which satisfies both conditions. By scaling this function by $s$ and translating by $b$, we can define wavelet functions $\psi_{s,b}$:

$$\psi_{s,b}(t) = \frac{1}{\sqrt{s}}\psi(\frac{t-b}{s})$$

Note that $s > 0$ is required and negative scales are undefined. The CWT of a signal, which decomposes $x$ as a set of coefficients defined at each scale $s$ and translation $b$, can then be written as:

$$W_x(s,b) = \int_{-\infty}^{\infty} \frac{1}{\sqrt{s}}\psi(\frac{t-b}{s})x(t)dt$$

Since $\psi$ has small local support, the integral (or sum) can be efficiently calculated. This transforms the signal $x$ from a one dimensional domain to a two dimensional domain of time and scales by convolution with a wavelet function at each scale $s$.

The scale parameters must be chosen according to prior knowledge about the problem and the input signal. Converting from frequencies to scales is one method of guiding the choice of scales. For example, a conversion from scales to frequencies can be estimated using the center frequency of the mother wavelet $F_c$, $F_s = \frac{F_c}{s}$ [10]. However, converting from scales to frequency is not useful unless prior knowledge about the signal is available or assumptions are made on the relevant frequency content of the signal. We are interested in the setting where this is not the case, i.e. the relevant frequency content of the signal is not known, and show how the scale parameters can be learned for a given problem using a combination of the wavelet transform and convolutional neural networks [22].

## 3 Wavelet deconvolutions

In this discussion, we focus on the context of machine learning problems on time series using neural networks. The setting is summarized as follows: we are given a set of data $x_1, x_2, ...x_n$ and targets $y_1, y_2, ...y_n$. Each $x_i$ is a (possibly multivariate) discrete time series signal that is preprocessed and passed to a neural network that learns to predict the target $y_i$. In many applications of interest, the neural network is a convolutional neural network and the preprocessing step takes the form of a decomposition of the signal into the time and frequency domain.

We replace the preprocessing step of converting the signal from the time domain to the time/frequency domain with a layer in the neural network. This layer, called the wavelet deconvolution (WD) layer, calculates the wavelet transform (WT) on an input signal in the forward pass, feeding the transformed signal to subsequent layers. It also computes the gradients of the loss function with respect to the scale parameters in the backward pass, allowing the network to adapt the transform to the problem it is tasked to solve.

The benefits of adding the WD layer as the first layer in a network include:

- Learning the relevant spectral content in the input with backpropagation
- Implicitly adapting the kernel support (filter size) via the scale parameter
- Reducing the number of hyperparameters to be optimized with expensive (in time and data) cross-validation procedures

Another benefit of this approach can be seen by considering the case of learning an optimal time/frequency decomposition of the signal using only CNN. Theoretically, a CNN could learn an optimal decomposition of the signal from the input in the time domain [11]. However, this would require careful selection of the correct filter sizes and costly data and training time. The WD layer circumvents these costs by fixing the form of the decomposition of the signal as the WT and learning the filter sizes. We note that any parametrized time-frequency decomposition of the signal can replace the WT in this method provided the parameters are differentiable with respect to the error. A further line of research could be relaxing the parameterization and allowing the layer to learn an empirical mode decomposition from the data such as the Hilbert Huang Transform [16], however we leave this as future work.

We now describe the details of the WD layer and show that the gradients of the scales can be calculated using backpropagation. The single-channel case is presented here but the extension to a multi-channel

signal is obtained by applying the transform to each channel. Given an input signal $x \in R^N$ with $N$ samples and a set of scales $s \in R^M$ with $s > 0$, the forward pass on the WD layer produces the signal $z \in R^{N \times M}$:

$$z_i = x * \psi_{s_i} \forall i = 1...M$$

We can equivalently write the convolution ($*$) as a summation:

$$z_{ij} = \sum_{k=1}^{K} \psi_{s_i,k} x_{j+k}$$

$$\text{for } i = 1...M \text{ and } j = 1...N$$

where $\psi_{s_i}$ is the wavelet function at scale $s_i$ discretized over a grid of $K$ points.

$$\psi_{s_i,t} = \frac{2}{\pi^{\frac{1}{4}} \sqrt{3s_i}} (\frac{t^2}{s_i^2} - 1) e^{-\frac{t^2}{s_i^2}}$$

$$t \in \{-\frac{K-1}{2}, ...0, ...\frac{K-1}{2}\}$$

The backward pass of the WD layer involves calculating $\delta E / \delta s_i$, where $E$ is the differentiable loss function being minimized by the network. Typically the loss function is the mean squared error or categorical cross entropy. Backpropagation on the layers following the WD layer yield $\delta E / \delta z_{ij}$, which is the gradient with respect to the output of the WD layer. We can write the partial derivative of $E$ with respect to each scale parameter $s_i$ as:

$$\frac{\delta E}{\delta s_i} = \sum_{k=1}^{K} \frac{\delta E}{\delta \psi_{s_i,k}} \frac{\delta \psi_{s_i,k}}{\delta s_i}$$

The gradient with respect to the filter $\psi_{s_i,k}$ can be written using $\delta E / \delta z_{ij}$:

$$\frac{\delta E}{\delta \psi_{s_i,k}} = \sum_{j=1}^{N} \frac{\delta E}{\delta z_{ij}} \frac{\delta z_{ij}}{\delta \psi_{s_i,k}} = \sum_{j=1}^{N} \frac{\delta E}{\delta z_{ij}} x_{j+k}$$

Defining $A, M, G$ and their partial derivatives as:

$$A = \frac{2}{\pi^{\frac{1}{4}} \sqrt{3s_i}}, \frac{\delta A}{\delta s_i} = -\frac{3}{\pi^{\frac{1}{4}}} (3s_i)^{-\frac{3}{2}} \quad M = (\frac{t_k^2}{s_i^2} - 1), \frac{\delta M}{\delta s_i} = -\frac{2t_k^2}{s_i^3} \quad G = e^{-\frac{t_k^2}{s_i^2}}, \frac{\delta G}{\delta s_i} = \frac{2t_k^2}{s_i^3} e^{-\frac{t_k^2}{s_i^2}}$$

Then the gradient of $\psi_{s_i,k} = AMG$ with respect to the scale $s_i$ is:

$$\frac{\delta \psi_{s_i,k}}{\delta s_i} = A(M \frac{\delta G}{\delta s_i} + G \frac{\delta M}{\delta s_i}) + MG \frac{\delta A}{\delta s_i}$$

Finally, we can write:

$$\frac{\delta E}{\delta s_i} = \sum_{k=1}^{K} [(\frac{4t_k^4}{s_i^4} - \frac{9t_k^2}{s_i^2} + 1) \frac{e^{-\frac{t_k^2}{s_i^2}}}{\pi^{\frac{1}{4}} \sqrt{3s_i^3}}] \sum_{j=1}^{N} \frac{\delta E}{\delta z_{ij}} x_{j+k}$$

The gradients of the loss with respect to the scale parameters, $\frac{\delta E}{\delta s_i}$, are used to update the scales with gradient descent steps:

$$s_i^{'} = s_i - \gamma \frac{\delta E}{\delta s_i}$$

where $\gamma$ is the learning rate of the optimizer. In order for the wavelet function $\psi_s$ to be defined we include the constraint $s_i > 0$ for $i = 1...M$.

It is not immediately clear how the scale parameters $s_1...s_M$ should be initialized for problems where the relevant frequency content is unknown. We show empirically that a random initialization of $s$ such that the whole space of possible frequencies can be explored suffices. This can be done by dividing the range of frequencies present in the signal into bins with the number of bins equal to the number of scales. The bins, or frequency bands, can be of equal size (the typical case) or variable size (depending on the prior knowledge of the signal). The bins are then converted to ranges in the scale domain from which the initial scales are randomly chosen.

An interesting feature of the WD layer is the flexibility with respect to the width of the filter ($K$). Since at small scales the support of $\psi$ is small as well, the width can be chosen to be small for some computational improvement. In fact, $K$ can be set dynamically according to the current value of the scale e.g. $K = \min(10s, N)$.

Another benefit to this approach is that a level of interpretability is added to the network. This can be achieved by examining the values of the scale parameters after training as they reveal the frequency content important to the classification task. In our experiments we show that the scale parameters converge to the true frequencies used to generate an artificial dataset.

## 4   Related work

Improving the performance of deep neural networks, particularly CNN, by using the network to learn features from close to the raw input has been proven to be a successful approach [15, 21, 28, 29, 34]. There are two main directions to this line of research, each with advantages and challenges.

One direction involves applying convolutional filters directly to raw input signals, assuming multiple layers of convolutions and max-pooling will be able to learn the appropriate features [32, 18, 3, 25, 40, 39]. While this is theoretically feasible, the network architecture selection and optimization become complicated as the number of layers is increased. Interpretability also suffers as the stack of features is difficult to interpret.

The other direction involves tuning the parameters of the preprocessing step by gradient descent. For example, using backpropagation to calculate the gradients of the mel-filter banks commonly used in automatic speech recognition. The gradients are then used to optimize the shape of the filters [31]. By jointly optimizing the feature extraction steps with the rest of the network, the feature extraction can be modified to be optimal for the classification task at hand. However, the drawback to this approach is that it requires a set of hand-crafted features with parameters that are differentiable with respect to the loss function. In addition, the shapes of the hand crafted filters are distorted by the gradients after many update steps since each point in the filter is updated independently. This causes the resulting filters to be uninterpretable and sometimes unstable [31].

Our work combines these two approaches by assuming a standard form for the feature extraction with provable qualities, i.e. the wavelet transform, and modifying the parameters of the transform using gradient descent. This combination simplifies the optimization process and circumvents the need for a pre-designed feature extraction step.

Wavelet coefficients and features extracted from wavelet coefficients have been used to train convolutional neural networks previously [23, 36, 19]. However, this work is the first, to the best of our knowledge, to optimize the scale parameters with gradient descent.

## 5   Results

Our experiments on artificial and real-world datasets show that including the wavelet deconvolution layer as the first layer of a neural network leads to improved accuracy as well as a reduction in tunable hyperparameters. Furthermore, we show that the learned scales in the WD layer converge

to the frequencies present in the signal, adding interpretability to the learned model. In all of our experiments, we implement and optimize the networks using the Tensorflow library [1] with the Keras wrapper [9]. [1]

## 5.1 Artificial data

We generate an artificial dataset to compare the performance of the WD layer to a CNN and verify that the learned scale parameters converge to the frequencies present in the signal. This dataset consists of randomly generated signals, with each signal containing three frequency components separated by time. The signals are separated into two classes based on the ordering of the frequency components; one class contains signals with components ordered from low frequency to high frequency while the other class contains signals with components ordered from high to low. Clearly, these two classes cannot be classified using a simple Fourier transform as the temporal order of the frequency components is important. Fig 2 shows examples from each class and demonstrates that they are indistinguishable using the Fourier transform. The purpose of this experiment and the design of this dataset is to show the WD layer can learn the spectral content relevant to a task.

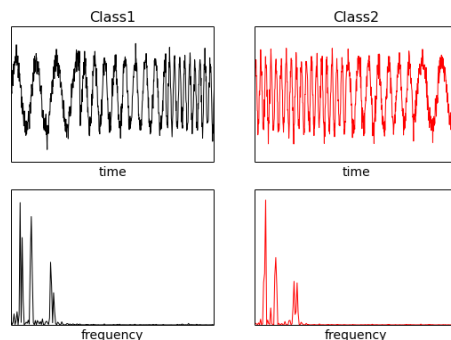

Figure 2: A positive (Class1) example and negative (Class2) example from the artificial two class dataset are shown in the first row. Examples from Class1 are generated by changing the frequency of the signal from low to high over time. Examples from Class2 are generated by changing the frequency of the signal from high to low over time. The plots in the second row show the Fourier Transform of the two examples. Without accounting for the frequency content over time, the two examples look identical.

We train two networks on examples from each class and compare their performance. The baseline network is a 4 layer CNN with Max-pooling [21] ending with a single unit for classification. The other network replaces the first layer with a WD layer while maintaining the same number of parameters. Both networks are optimized with Adam [20] using a fixed learning rate of 0.001 and a batch size of 4. Fig 3 shows the learning dynamics of both of these networks on this problem as well as a comparison of their performance. The network with the WD layer learns much faster than the CNN thanks to its flexibility in learning filter size and scales as well as achieving a near perfect AUC-ROC score.

Additionally, we observed that the scale parameters learned by the WD network converged to the frequencies of the signal components. We experimented with different initializations of the scale parameters to verify this behavior was consistent. This is shown in the third panel in Fig 3.

## 5.2 TIMIT

The TIMIT dataset is a well-known phone recognition benchmark dataset for automatic speech recognition (ASR). Performance on the TIMIT dataset has steadily improved over the years due to many iterations in engineering features for the phone recognition problem. Prior to the rebirth of deep neural networks in ASR, the community converged to HMM-GMM trained on features based on mel-filters applied to the speech signal and transformed using a discrete cosine transform (DCT) [41]. More recent work improves performance by omitting the DCT step and applies CNN directly to

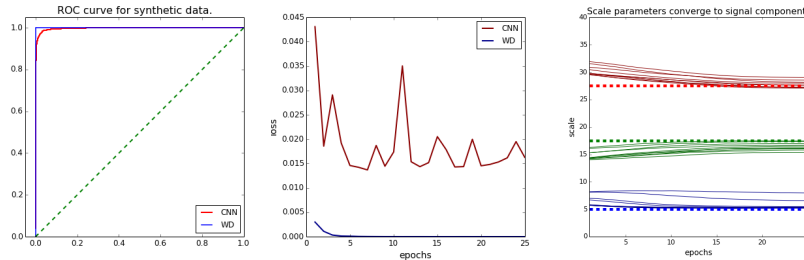

Figure 3: Top left: ROC curve of the WD network (blue) and the convolutional network (red) on the synthetic dataset. The WD network achieves a perfect score on the test dataset. Top right: Validation loss over training epochs for the WD network (blue) and the convolutional network (red). The WD network learns much faster than the CNN and achieves a lower loss value overall. Bottom: The width parameters of the WD network over training epochs (solid lines) from several random initializations alongside the true frequency used to generate the synthetic dataset (dashed lines). We observed that the width parameters converged to the true frequency components indicating that the WD layer is able to uncover the relevant frequency information for a problem.

Table 1: best reported PER on the Timit dataset without context dependence

| Method | PER (Phone Error Rate) |
| --- | --- |
| DNN with ReLU units [37] | 20.8 |
| DNN + RNN [12] | 18.8 |
| CNN [38] | 18.9 |
| **WD + CNN (this work)** | 18.1 |
| LSTM RNN [13] | 17.7 |
| Hierarchical CNN [38] | **16.5** |

the output of the mel-filter banks [15, 2, 17, 28]. Our goal is to extend this further by removing the mel-filter banks and attempting to learn the appropriate filters using the WD layer. Clearly, this is a difficult task as the mel-filter banks represent the result of decades of research and experimentation. Our motivation is to show that the WD layer is adaptable to different problem spaces and provide an approach that circumvents the need for extensive feature engineering.

To ensure a fair comparison to previous results, we replicated the non-hierarchical CNN given in [38], a 4 layer network with 1 convolutional layer followed by 3 fully connected layers. In our network, we remove the mel-filter bank preprocessing steps and added the WD layer as the first layer, using the speech signal directly as input to the WD layer. The WD layer passes the wavelet transform of the signal along with the $\Delta$ and $\Delta\Delta$ values to the CNN for classification in the forward pass. In the backward pass, the gradients of the scale parameters are calculated using the gradients from the CNN. We also use the optimization parameters presented in [38], except we replace SGD with the Adam [20] optimizer. A minibatch size of 100 was used. The learning rate was set to 0.001 initially and halved when the validation error plateaued. The dataset in its benchmark form consists of 3696 spoken sentences used for training and 192 sentences for testing. The sentences are segmented into frames and labeled by phones from a set of 39 different phones. 10% of the 3696 training dataset are used as a validation set for hyperparameters optimization and early stopping for regularization. The standard set of features used by CNN-based approaches consists of 41 mel-filter bank features extracted from the signal along with their $\Delta$ and $\Delta\Delta$ change values. The results using this network are shown in Table 1 alongside other strong performing methods.

Although the WD layer does not outperform the best CNN based approach using the hand-crafted signal decomposition, it is clear that the approach is competitive achieving a close 18.1% PER despite removing all of the engineered features. This is expected as the mel-filter bank based decomposition is well suited to this speech recognition task. Removing the mel-filter bank features puts the WD+CNN

model at a significant disadvantage compared to the other methods because the model must first learn the appropriate preprocessing steps.

However, these results show that with minimal engineering effort and a reduction in tunable hyper-parameters the WD layer offers an effective alternative to the mel filter bank features. Introducing the WD layer to the CNN eliminated 7 tunable hyperparameters from the preprocessing step of the baseline CNN. This is significant as it shows the WD layer can learn a set of features equivalent to a carefully crafted feature extraction process directly from the data. By plotting the frequency response of the learned wavelets, as shown in Fig 4, we observe that they resemble the triangular filters of the mel-filter bank.

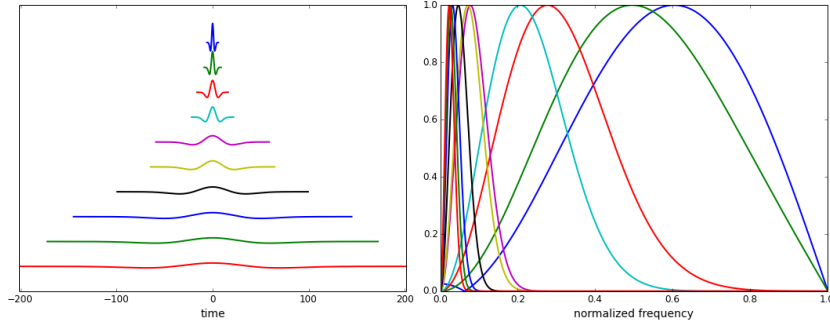

Figure 4: Visualization of 10 learned wavelet filters from the network trained on the TIMIT dataset. The left pane shows the scaled wavelet functions and the right pane shows the frequency response of each of the learned wavelet filters. The learned wavelet filters closely resemble the triangular mel filter banks commonly used in automatic speech recognition systems both in shape and spacing across the frequency spectrum.

## 5.3   UCR Haptics dataset

In this set of experiments, we evaluate the performance of the WD layer on a dataset for which an engineered preprocessing stage is not known. As shown in Table 2, methods using hand crafted features perform poorly on the dataset.

The Haptics dataset is part of the UCR Time Series Classification dataset [8]. The data is comprised of time series of length 1092 divided into 5 classes. The data is provided mean-centered and normalized and a training/testing split of the data is set. We further split the training data with 10-fold crossvalidation for early stopping and selecting number of filters. We train the following 7 layer network on the dataset: WD layer with 7 scales, three convolutional+maxpooling layers with 32 filters each and 2x5 kernels (first dimension is scales and second dimension is time), three fully connected layers with 100, 25, and 5 units. Dropout [35] with $p = 0.3$ was added after every layer for regularization. The nonlinear activations after every layer were ReLU [26]. The network was trained using Adam [20] with default parameters: $lr = 0.001$, $\beta_1 = 0.9$, and $\beta_2 = 0.999$. The network was trained for 1000 epochs with a batch size of 2 and the weights with the best validation loss were saved.

The results in Table 2 show that the WD layer achieves the best performance with an error of 0.425, improving the next best performing method by an absolue 2.4%. The second best performing method, the Fully Convolutional Network (FCN) [40], is a network of 1-D convolutional units that also does not require any preprocessing or feature extraction steps and has a similar number of parameters to our method. Other methods such as Dynamic Time Warping [5], COTE [6], and BOSS [33] depend on feature extraction steps which may not be suitable to this task. We believe the improvement shown here, especially with respect to other CNN based methods with similar model complexities, shows the WD layer learns a spectral decomposition of the time series which results in improved classification accuracy.

Table 2: Testing error on the Haptics dataset

| Method | Test Error |
|--------|:----------:|
| DTW [5] | 0.623 |
| BOSS [33] | 0.536 |
| ResNet [40] | 0.495 |
| COTE [6] | 0.488 |
| FCN [40] | 0.449 |
| **WD + CNN (this work)** | **0.425** |

## 6   Discussion

We demonstrate that the WD layer provides a powerful and flexible approach to learning the parameters of the spectral decomposition of a signal. Combined with the backpropagation algorithm for calculating gradients with respect to a loss function, the WD layer can automatically set the filter widths to maximize classification accuracy. Although any parameterized transform can be used, there are two benefits to using the wavelet transform that are not realized by other transforms. Firstly, the wavelet functions are differentiable with respect to the scales which allows optimization with the backpropagation algorithm. Secondly, the scale parameters control both the target frequency as well as the filter width allowing a multiscale decomposition of the signal within a single layer of the network.

One challenge to the optimization of the WD layer using stochastic gradient descent (SGD) with a fixed learning rate is that the scale parameters can change too slowly relative to their magnitude and convergence can be slow. This is caused by the multiscale feature of the wavelet transform. When the magnitude of the scale parameter is small, small changes to the parameters can capture change in high frequencies effectively. At lower frequencies when the magnitude of the scale parameter is large, many steps are required. Fortunately, more advanced optimization techniques with variable and per-parameter learning rates, such as Adam [20] and Adadelta [42], circumvent this problem. We found that using Adam (a standard choice for deep neural networks) with the default parameters greatly sped up training over using SGD with a fixed learning rate. Thus, this method requires a variable learning rate in order to effectively learn the scale parameters.

## 7   Conclusion

In this paper, we used the wavelet transform and convolutional neural networks to learn the parameters of a spectral decomposition for classification of signals. By learning the wavelet scales of the wavelet transform with backpropagation and gradient descent, we avoid having to choose the parameters of the spectral decomposition using cross-validation. We showed that the decomposition learned by backpropagation equaled or outperformed hand-selected spectral decompositions. In addition, the learned scale parameters reveal the frequency content of the signal important to the classification task, adding a layer of interpretability to the deep neural network. As future work, we plan to investigate how to extend the WD layer to signals in higher dimensions, such as images and video, as well as generalizing the wavelet transform to empirical mode decompositions.

### Acknowledgments

This work was supported in part by NSF Award #1302231.

## Footnotes

[1]Code implementing the WD layer can be found at `https://github.com/haidark/WaveletDeconv`

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
