[Reviews · NeurIPS 2018]

Reviewer 1



In the context of time series classification, this paper introduces the Wavelet Deconvolution layer, which is able to learn the filters width of a neural network-based classifier, which is usually tuned by hand through hyper-parameters grid search. Experimental studies on time series are presented, including a pilot study using artificial data, a phone recognition study on TIMIT and a time series classification study on the Haptic UCR dataset. It is shown that the proposed method leads to similar of better results than state-of-the-art systems. Quality: The paper is correct to my understanding. The experiments are convincing and clearly show the benefit of the proposed approach. Having two different studies on different tasks is very good, showing the generalization capability of the approach on various cases. Minor comment: in section 5.2, when describing the use of Mel filterbank as input to a CNN, [29] does not use filterbank as input but temporal raw speech. There is also several papers using raw speech as input which have studied the learned filters of a CNN [1][2][3], the authors could discuss that when presenting the frequency responses of the wavelets at the end of section 5.2. Clarity: the paper is well-written and easy to follow. Originality: The proposed method is novel and an original application of the wavelet transform. Significance: This work is definitely significant. Being able to learn several hyper-parameters of a CNN instead of running hundreds of experiments is very important, specially for tasks with no established preprocessing step. The interpretability of the wavelet scales is also a nice addition. The proposed WD layer could be widely adopted. Overall, this paper is significant and it should accepted for publication. References: [1] T. N. Sainath, R. J. Weiss, A. Senior, K. W. Wilson, and O. Vinyals. "Learning the Speech Front-end With Raw Waveform CLDNNs". Proceedings of the Annual Conference of the International Speech Communication Association (INTERSPEECH), 2015 [2] Z. Tüske, P. Golik, R. Schlüter, and H. Ney. "Acoustic Modeling with Deep Neural Networks Using Raw Time Signal for LVCSR". In Proceedings of the Annual Conference of the International Speech Communication Association (INTERSPEECH), pages 890–894, Singapore, September 2014. [3] D. Palaz, M. Magimai-Doss, and R. Collobert. "Analysis of CNN-Based Speech Recognition System Using Raw Speech as Input." Sixteenth Annual Conference of the International Speech Communication Association. 2015. UPDATE: The authors response clarified most of the questions raised with the paper, especialy the choice of presenting the method using a specific wavelet and the use of the "deconvolution" term, as well as experimental details. Hence my score remains the same: this paper is signiicant and should be accepted.

Reviewer 2



* Summary: This paper introduces a wavelet deconvolution (WD) layer for networks working on time-series. The WD performs a linear convolution with parametrized filters, using a closed-form formula which only involves one parameter per filter, the scale. The scale parameter varies continuously and can be adapted by gradient descent, removing the need for a preprocessing with e.g. mel filterbanks. A synthetic experiment on a toy classification example with time-frequency content shows that this layer improves results with respect to a CNN working directly on the raw waveform. Classification results on speech (TIMIT) show that accuracy is improved for a similar CNN architecture when the hand-crafted preprocessing is replaced by the WD layer, yielding results not too far from the state-of-the-art. Classification results on a very small dataset (UCR Haptics, 10x smaller than TIMIT) yield state-of-the-art classification results. * General opinion about the paper: This paper is quite well written, up to minor remarks. The general idea is clear. This paper is in the same vein as other recent papers which try to bridge the gap between hand-crafted features and completely black-box methods on raw data (see e.g. [Learning filterbanks from Raw speech for Phone Recognition, Zeghidour et al, ICASSP 2018]), except that it uses closed-form filters with few parameters, which allows to improve the regularity of the learnt filterbank. The authors could have more insisted on the potential impact of this method for dealing with very small data. The experiments support the claims of the authors, even if a deeper analysis in some parts would have been appreciated. The experiments are almost reproducible by reading the paper (cf remarks below). The main criticism to me lies in the lack of generality of the paper, which limits its significance. As mentioned in the conclusion by the authors, the method would be applicable to many different kinds of parametrized time-frequency transforms, as long as the parameters vary continuously (cf remarks below): I would have appreciated such a general formulation. The Mexican hat wavelets could still have been used as an illustration. Also, the name "wavelet deconvolution" appears very ill-chosen to me, insofar as the layer precisely performs a convolution with a wavelet filterbank; this has nothing to do with the deconvolution problem $y = A \ast x$ where x is unknown. * Detailed remarks: 1) Line 63, $F_s = F_c / (s \Delta)$. What is $\Delta$? It is not introduced. 2) It would be nice to give in 1 sentence what the straightforward extension to the multi-channel signal (l. 95-96) would be, because it is not obvious to me. Does it consist in applying this transform to each channel separately? 3) If I understood correctly, the WD layer performs a linear operation. Spectrograms are typically computed after a non-linear transform z_i = |x \star h_i| where the h_i are the different filters of a local Fourier family (possibly on the mel scale). Despite this difference, the two operations are very often compared in the paper (cf Figure 1 for instance). Why not including the non-linearity here? 4) What motivates the choice of the Mexican Hat wavelets? Indeed, as it is real and symmetric, it cannot be analytical in Fourier: this makes it very different from the standard mel filterbanks used in audio. 5) I think that the arguments supporting the use of the wavelet transform (l.257-262) could be applied to other transforms, as long as they have continuous parameters. It is true that the wavelet parametrization in time-frequency is already good for many applications, but allowing more flexibility with more parameters would be useful in many cases. 6) The derivation of the gradient l.106-111 is not essential to the paper, especially with the use of automatic differentiation software. I would recommend to leave it in the appendix in order to leave more space for the rest of the paper. 7) In Figure 2, right panel, the scale parameters seem to be already very close to the value which is chosen. How is this possible with a random initialization? 8) The nature of the initialization used (l.118) would require a clarification. Are the frequencies of the wavelets F_s evenly spaced (and the scales s adapted to match this spacing)? Or are the scales s themselves evenly spaced? 9) In the TIMIT experiment, why is it necessary to add the $\delta$ and $\delta\delta$ values of the WD to the network? What was the performance without it? How many filters are used and how does it compare to the number of mel filterbanks features used in the other methods? This hyperparameter value is missing (or is there a typo in the caption of Figure 2: "of (the) 10 filters"?). An ablation experiment would also be interesting for this dataset: what would be the performance of the network with a random initialization of the filters, without tuning their scales? 10) Figure 2 is not that illuminating. Since the use of this layer is the alleged interpretability of the layer, it would have been nice to simply show the set of all scales (possibly sorted). Do they follow a geometric progression? Or an arithmetic one? Further, the comment related to the shape of the filters in the caption is slightly misleading: the shape of the filters is defined by the choce of the Mexican hat wavelet (which is then dilated), it has nothing to do with the training. 11) In the UCR experiment, how does the proposed method compare to the others in terms of numbers of parameters and training time? 12) The first paragraph of the conclusion should be moved somewhere else. 13) I think that there is a minor typo in the Equation after line 98: x_{j+k} should be replaced with x_{j-k}.

Reviewer 3



In this paper the authors present parameterized convolution layer. The parameterization is done using the wavelet transform whose parameters are learnt using gradient descent. The benefits of this parameterization are that the convergence is faster and minimal hyperparameter tuning is necessary either for initialization of the CNN layer parameters or the pre-processing steps. The authors report improved performance using the proposed method compared to CNNs with hand-crafted features in 2 different tasks. Compared to previous approaches which either learn the temporal convolution layers from random initialization a major purported advantage of the current approach is interpretability of the wavelet transforms. In the paper the authors compare with CNNs which operate on log mel features but not with CNNs which operate on raw waveforms. Despite the advantages of the proposed method compared to the raw waveform CNNs in terms of computational complexity, interpretability and stability during optimization it would be interesting to see this comparison in terms of performance. The authors describe the challenges in optimization with vanilla SGD and uniform learning rate across all parameters and the significant advantage due to use of techniques like Adam. It would be informative to expand this section and present a more detailed analysis. In figure 3 it is not clear how many wavelets were used in the WD layer and how this specific subset of 10 were chosen ?